# Effects of Nutritional Restriction during Laying Period of Fat and Lean Line Broiler Breeder Hens on Meat Quality Traits of Offspring

**DOI:** 10.3390/ani11082434

**Published:** 2021-08-18

**Authors:** Feng Li, Yingjie Xie, Xue Yang, Yuanyuan Zhang, Baojing Cheng, Anshan Shan

**Affiliations:** Institute of Animal Nutrition, Northeast Agricultural University, Harbin 150030, China; lifeng@neau.edu.cn (F.L.); xieyingjie@neau.edu.cn (Y.X.); yangxue01@caas.cn (X.Y.); zhangyuanyuan@hrfri.ac.cn (Y.Z.); chengbaojing@neau.edu.cn (B.C.)

**Keywords:** broiler, feed intake, maternal effect, offspring, meat quality, muscle morphology

## Abstract

**Simple Summary:**

The meat quality of livestock products is widely appreciated. Maternal nutrition can affect the deposition of nutrients in eggs, and then change the apparent metabolism, development process, and performance of offspring. Our research indicated that meat quality traits are also affected by maternal nutritional level and are related to the nutritional requirements of different genotypes. Some of the effects disappeared at the end of the growth stage. These situations remind poultry producers to consider the impact of feed restrictions on the quality of meat for future generations.

**Abstract:**

The offspring meat quality of hens undergoing a 25% dietary restriction treatment during the laying period were evaluated in fat and lean line breeder. A total of 768 female birds (384/line) were randomly assigned to four groups (12 replicates/group, 16 birds/replicates). Maternal feed restriction (MFR) and normal started at 27 weeks of age. Offspring broilers were fed ad libitum. The offspring meat quality traits and muscle fiber morphology in different periods were measured. At birth, significant interactions were found on breast muscle fiber morphology (*p* < 0.05). At 28 days, MFR decreased breast water content and increased thigh crude fat content, and significant interactions were observed on breast crude fat and protein contents (*p* < 0.05). At 56 days, MFR affected morphology of peroneus longus muscle tissue, and significant interactions were found on thigh redness at 48 h and amino acid contents in breast and thigh muscle (*p* < 0.05). Overall, MRF may lead to offspring birth sarcopenia. Such offspring grow more easily to deposit fat in a nutritious environment, but they will self-regulate adverse symptoms during growth and development. The two lines respond differently to maternal nutritional disturbance due to different nutritional requirements and metabolic patterns.

## 1. Introduction

According to statistics, poultry has become the world’s largest consumption of animal protein with a per capita consumption of 14.88 kilograms in 2020 [1]. With the emergence of picky consumers, high-quality meat is pursued.

In recent decades, great progress has been made in the genetic selection of fast-growing broilers [2], but the meat quality has decreased significantly with the excessive deposition of fat (especially abdominal fat). Selection and breeding of low-fat broilers were designed to improve the excessive deposition of abdominal fat in rapid-growth broilers. Lean line (L) and fat line (F) broiler breeds were successfully established by selecting for long-term differences in abdominal fat percentage (AFP) and for plasma, very low-density lipoprotein (VLDL) concentration. Due to the different genotypes, the two lines of broiler breeds showed obvious differences in meat quality, including meat color, tenderness, chemical content, and so on [3]. Most importantly, L broilers have higher intramuscular fat content than F, which improved the taste of meat [4].

With the prosperity of the broiler industry, researchers gradually pay attention to the role of “maternal effects” [5]. The broiler breeder hens are usually forced to live in restricted-feeding breeding regimes to prevent obesity and lower laying rates [6], which has serious maternal effects on their offspring. Feeding restriction reduces egg weight and alters the chemical composition of eggs [7]. Subsequently, the birthweight of broiler offspring decreased, and blood biochemical indexes and hormone levels were changed [8]. Although many studies have linked maternal nutritional status to phenotypic changes in body weight and growth of offspring, there are few investigations into meat quality.

For this reason, broiler hens were treated with two different daily feed intakes during laying period, and the effects of this treatment on offspring meat quality, muscle composition, and muscle fiber morphology were investigated. Our prediction is that low feeding treatment will have a significant effect on the meat quality of broiler offspring, and the two strains will show their own characteristics.

## 2. Materials and Methods

The Handbook of Modern Broiler Production [9] was a guideline for animal feeding and handling procedures during experiments.

### 2.1. Experimental Design and Management of Hens

The F and L broilers of the Northeast Agricultural University (Harbin, China) were used in this experiment. The two lines were selected from 1996 based on AFP and plasma VLDL levels [10]. In the growing period, all broiler breeder hens were given the same diet (0–6 weeks: 12.10 MJ kg^−1^ ME, 179.0 g kg^−1^ CP; 7–18 weeks: 11.82 MJ kg^−1^ ME, 147.0 g kg^−1^ CP; 19–24 weeks: 11.61 MJ kg^−1^ ME, 161.0 g kg^−1^ CP). At 23 weeks, in total 768 hens (384/line, no significant difference in body weight) were assigned to 4 treatments, with 12 replicates of 16 birds per treatment. The four treatments were: LN, L with normal daily feed intake; LR, L with restricted daily feed intake; FN, F with normal daily feed intake; FR, F with restricted daily feed intake. A repetition was composed of eight adjacent metal cages with two hens per cage. The different replicate treatments were evenly distributed at different heights to reduce spatial variation. Laying diets for hens (See Table 1) were fed to all groups after 24 weeks. When laying rate reached 5% (27 weeks), broiler breeder hens in LN and FN groups were fed with nutrition recommendation of Chinese Broilers Feeding Standard (27–28 weeks: 1050 g/bird; 29–32 weeks: 1190 g/bird; 33–42 weeks: 1169 g/bird; 43-57 weeks: 1141 g/bird; 58-weeks 1064 g/bird) [11], and those in LR and FR groups were fed with 75% of nutrition recommendation (27–28 weeks: 787.5 g/bird; 29–32 weeks: 892.5 g/bird; 33–42 weeks: 876.75 g/bird; 43–57 weeks: 855.75 g/bird; 58-weeks 798 g/bird). Feed was given at 08:00 and 16:00, and drinking water was supplied ad libitum. The light was a combination of natural and artificial light. At 23 weeks, the photoperiod using artificial light was 13 h day^−1^ and gradually increased at a rate of 0.5 h week^−1^ up to 16 h day^−1^. The temperature inside the house was maintained at about 20 °C.

At 40 weeks, 35 males of each line were used to obtain fertile eggs by artificial insemination. Artificial insemination was conducted every 5 days from 2:00 to 5:00 in the afternoon. The depth of insemination was 2–3 cm, and the amount of insemination was 0.025–0.03 mL. A total of 1500 settable eggs were collected from each treatment and placed in an automatic incubator (FT-ZF 10, Chun Ming Fang Tong Electronic Co., LTD, Beijing, China). Each egg was put in a string bag and marked with its treatment and replicate. During incubation, the blunt tips of all eggs were upward. The temperature was 38.4 °C on days 1–6, 38.1 °C on days 7–12, 37.2 °C on days 13–18, and 36.9 °C on days 19–21. The humidity was between 65% and 75%.

### 2.2. Experimental Design and Management of Offspring Broilers

The grouping and repetition of offspring broilers were strictly consistent with the de-sign of the hen trial. There were 4 groups with 12 replicates in each group and 20 offspring broilers in each replicate randomly selected from the same replicate of hen trial. Offspring broilers were reared for eight weeks. Corn-soybean-based diets (See Table 1) and water were provided ad libitum. The temperature in the hennery was 34–36 °C for the first week and then gradually decreased to 21–23 °C at a rate of 2 °C week^−1^. Humidity control was approximately 65%.

### 2.3. Data and Sample Collection

At days 1, 28, and 56, two offspring birds from each replicate were selected to sacrifice. Breast muscles and thigh muscles were taken out for further measurements. Superficial fat and connective tissues were trimmed from fillets, which were subsequently stored at 2 to 4 °C throughout handling and measurements.

Meat quality traits were measured using breast muscles and thigh muscles of 56-day-age offspring. The color parameters, lightness (L*), redness (a*), and yellowness (b*), were measured on breast and thigh muscle using an automatic color difference meter (TC-P11G, Beijing Optical Instrument Factory, Beijing, China) and averaged at 6 to 9 sites of each meat sample at 2, 24, and 48 h after sacrifice. The pH values were measured in left breast and thigh muscles with a portable digital display pH meter (SFK Meat Systems, Kolding, Denmark) at 45 min and 24 h after sacrifice. Drip loss, cook loss, and the Warner-Bratzler shear force were measured using right breast muscle. The meat samples were cut to 2 × 2 × 1 cm for drip loss measurement. The samples were threaded by nylon thread, suspended in a plastic cup, hung for 24 h at 4 °C, after which the surface juice was blotted with filter paper. Cooking loss samples were treated as follows: the meat samples were placed in plastic bags to age, at 15–16 °C for 24 h and then at 4 °C for 24 h, after which they were placed at room temperature for 1 h. A thermometer was inserted into the center of the sample. The meat samples were matured in a water bath at 80 °C, keeping the plastic bag open upward, until the internal temperature reached 70 °C, after which they were cooled to room temperature and dried with filter paper. After cooking loss measurement, meat samples were cut to 1.27 cm diameter × 1.5 cm height size to measure Warner-Bratzler shear force using a universal Warner-Bratzler tester (C-LM3, Northeast Agricultural University, Harbin, China). In addition, press loss was measured using a dilatometer (WZ-2, Nanjing Soil Instrument Factory Co. LTD, Nanjing, China) with 5 cm^2^ round samples cut with a circular sampler. The circle was sandwiched between two layers of gauze, with 18 layers of qualitative filter paper on each side, and then maintained at a pressure of 35 kg for 5 min.

The moisture content, crude protein content, crude fat content, and amino acid content were determined using fresh left breast and thigh muscles of 28- and 56-day-old offspring. Moisture content was measured by oven drying method, 5–8 g muscle tissue was dried at 103 ± 2 °C for 4–5 h. Crude protein and fat content was measured using the air-dried sample. Crude protein was determined by using Kjeldahl method [12] at Kjeldahl apparatus (KT-2300, FOSS, Hilleroed, Denmark). Crude fat content was determined by using the Diethylether Extraction-Submersion Method (Soxhlet extractor method) [12]. The amino acid level was determined by the hydrochloric acid hydrolysis method of 60–120 mg fresh breast muscle or thigh muscle [13] in an amino acid analyzer (L-8800 Hitachi Automatic Amino Acid Analyzer, Tokyo, Japan).

The morphology of the muscle tissue was determined with 1- and 56-day-old breast and thigh muscles. For 1-day-old, whole thigh muscles, not distinguished into different muscle parts, were assayed. At 56 days of age, gastrocnemius and longus muscles were assayed separately. Each muscle sample was fixed in 10% (*v*/*v*) buffered formalin fixative for 24 h and paraffin-embedded, and a microtome (RM2235, Leica Microsystems Inc., Buffalo Grove, Wetzlar, German) was used to prepare 6-μm-thick sections. The next step consisted of paraffin removal and slide hydration, using xylene, and different concentrations of alcohol and water. Samples were stained following the hematoxylin and eosin staining protocol. Samples were then dehydrated again and mounted. The camera microscope was used to collect and save the still image. Each section randomly selected 8–12 visual fields, the number of muscle fibers in each field was not less than 100. Images of each paraffin section were analyzed with Image Analysis System (Motic Images Advanced, Motic China Group Co., Ltd., Shenzhen, China) to calculate the fiber diameter and density.

### 2.4. Statistical Analysis

All data from this experiment were analyzed using a proc mixed statistical analysis system (SAS Inst. Inc., Cary, NC, USA) as a complete randomized block. Dietary intake and lines were fixed, and blocks were random. Specifically, 384 hens of each line were classified into 12 blocks based on weight at 23 weeks, and 32 hens with similar weight in one block were randomly assigned to the treatments of normal daily feed intake and restricted daily feed intake as a replicate. The following model was used to analyze these data: Yijk = μ + αi + βj + (αβ)ij + Pk + εijk. Yijk is the value of the individual sample from each replicate; μ is the overall mean; αi is the dietary intake effect; βj is the line effect; (αβ)ij is the interaction between dietary intake and line; Pk is the effect of block; and εijk is the error component. When differences in the treatment means were found by ANOVA, Duncan’s multiple range test was applied to separate means. Values of *p* ≤ 0.05 were significant.

## 3. Results

The color parameters of 56-day-old offspring breast and thigh muscle are shown in Table 2 and Table 3. The L* of breast and thigh muscles of 56-day-old L offspring at 48 h was significantly lower than that of F (*p* < 0.05). The a* of breast muscle of 56-day-old L offspring at 2, 24, and 48 h was significantly higher than that of F (*p* < 0.05). There was a significant interaction between line and intake on the a* of thigh muscle of offspring at 48 h (*p* < 0.05), which was decreased by RI in the F, but not in the L. No significant effects of feed intake of laying hens were discovered on the color parameters of offspring chest muscle. The results of pH value, press loss, drip loss, cooking loss, and shear force are presented in Table 4. Only drip loss of breast muscle of 56-day-old L offspring was significantly higher than that of F (*p* < 0.05).

The water, crude fat, and crude protein content of 28- and 56-day-old offspring breast and thigh muscle are shown in Table 5. In 28-day-old offspring breast muscle, RI of hens during the laying period decreased the water content (*p* < 0.05). A significant interaction between line and feed intake were found on the crude fat and crude protein of air-dried breast muscle, in which RI increased the crude fat content in F offspring and the crude protein content in L offspring (*p* < 0.05). In 28-day-old offspring thigh muscle, RI of hens during the laying period increased the crude fat content of air-dried muscle (*p* < 0.05). No significant effect was found on water, crude fat, and crude protein content of 56-day-old offspring breast and thigh muscle.

The amino acids contents of 56-day-old offspring breast and thigh muscle are shown in Table 6 and Table 7. In 56-day-old breast muscle, Methionine (Met), Leucine (Leu), and Phenylalanine (Phe) content in L offspring were significantly lower than that in F offspring, and Tyrosine (Tyr) content was on the contrary (*p* < 0.05). There was a significant interaction between line and feed intake on the Tyr level of breast muscle of 56-day-old offspring (*p* < 0.001), that is, RI decreased this index in F offspring, but increased in L offspring. In 56-day-old thigh muscle, Tyr content in L offspring was significantly higher than that in the F (*p* < 0.05). There were significant interactions between line and feed intake on practically all amino acids content (*p* < 0.05), except Cysteine (Cys), Proline (Pro), Met, and Histidine (His). RI significantly decreased the Valine (Val), Isoleucine (Ile), Leu, Tyr, and Phe contents in F thigh muscle, but increased the Threonine (Thr), Serine (Ser), Glutamic acid (Glu), Glycine (Gly), Alanine (Ala), Val, Ile, Leu, Tyr, Phe, and Arginine (Arg) contents in L thigh muscle.

The diameter and density of 1- and 56-day-old breast and thigh myofiber were shown in Table 8. Significant interactions between line and maternal feed intake were found on 1-day-old offspring breast muscle (*p* < 0.05). Fiber density of gastrocnemius muscle of 56-day-old L offspring was significantly lower than that of the F (*p* < 0.05). RI of hens during the laying period significantly increased the fiber diameter and decreased the fiber density in the peroneus longus muscle of 56-day-old offspring (*p* < 0.05).

## 4. Discussion

The environmental factors experienced by breeders during their breeding, whether positive or negative, can have a significant impact on the production efficiency and health of their offspring [14]. It is generally accepted that the prenatal environment is the most important “maternal effect” phase [5]. In this experiment, hens were restricted to feed at the beginning of the laying period and were treated for over ten weeks before the egg-collection time. Maternal undernutrition triggered breeders to relocate more energy towards growth and maintenance, but less towards reproduction [15,16]. Consequently, there are changes in the ovo composition and a reduction in the amount of energy available for progeny embryonic development [7]. Our previous research showed that nutritional restriction reduced the birth weight of offspring, but had no effect on the weight at day 56, daily gain, and feed conversion rate (FCR) [8]. A decreasing trend was found in breast muscle rate at day 28 and thigh muscle rate at day 56 [8]. Combined with recent data, the following inference is explained in this study.

Skeletal muscles are given a lower priority than vital organs such as the brain and heart during embryonic development, and inadequate nutrition directly affects the development of skeletal muscles of the offspring [17]. Skeletal muscle underdevelopment at birth is also the main cause of lower birth weight in offspring [18]. In the present study, after maternal RI treatment, offspring myofiber diameter and density appeared slightly different at birth in the two lines. This may result from the difference in the amount of energy required for the two lines of hens to maintain themselves and for their offspring during embryonic development. Undernutrition may lead to insufficient fiber development and a decrease in the number of muscle fibers [19,20]. The phenomenon in this study was that breast muscle rate at day 28 and thigh muscle rate at day 56 had a decreasing trend [8]. Restricted feeding may affect the embryonic development of the visceral organs of offspring [21], but it is not clear in this experiment. The size and metabolic activities of visceral organs and muscles will affect broiler growth and development [22].

Food enrichment in the subsequent environment of the offspring resulted in a compensatory growth trajectory and produces deposition of fat [23,24]. As all offspring raised in a nutrient-rich environment, the broiler with insufficient amount of muscle fiber reaches the nutrient intake needed for muscle maturation earlier. At 28 days, A lower water content in offspring breast muscle may indicate high developmental maturity of muscles. At that time, the offspring abdominal fat rate did not increase [8] and the crude fat content in the F thigh and breast muscles and L thigh muscles increased. Intramuscular fat was positively correlated with body fat [25]. This may indicate that excessive energy was converted into adipose development and lipid accumulation, which will increase the overall fatness of offspring [26]. The regulation of growth, development, and differentiation of adipose tissue and skeletal muscle is controlled by various endocrine, paracrine, and autocrine interactions, and there are interactions between adipose tissue and muscle cells [27]. Although the increase in muscle fat can bring better flavor [28], obesity may cause metabolic disorders and affect muscle protein metabolism to varying degrees [29,30].

Permanent environmental maternal variation accounts for a progressively smaller proportion of the phenotypic variation as offspring age increases [31]. The rate of fat deposition was significantly slower in the progeny over time [32]. In this experiment, no difference was observed in the crude fat and crude protein content in breast muscle at the end of growth. Compensatory growth in offspring broilers showed that chicks from restricted mothers reached adult age earlier than those from freely growing mothers [24]. It is not clear whether maternal nutrition alters the maturational age of broilers, which at 56 days of age may have prematurely senesced to produce altered amino acid content caused by changes in amino acid metabolism [33,34].

The meat quality of poultry is greatly influenced by heredity [2,35]. Long term selection for AFP and plasma VLDL levels yielded two lines with distinct genotypes [36]. An et al. (2017) [3] described in detail the differences in meat quality traits between the two lines. In this experiment, there were significant differences in meat quality between the two lines, such as meat color, dripping loss, amino acids levels in muscle, etc.

In the above, we have shown that the differences in part result from different nutritional dependence of lines. There are differences in the nutritional requirements and reproductive performance of broilers between the two genotypes [34,37]. When F hens are restricted by feed, the hens are subject to more nutritional stress and cause their offspring to develop muscle weakness at birth. Muscle development was completed early during the growth and development of the offspring, and then the condition of overnutrition manifests as the accumulation of intramuscular fat (perhaps body and abdominal fat). Hu et al. (2010) [38] similarly showed that the effect of feed intake restriction in hens on fat deposition in offspring varies between lines. The different degree of fat deposition in two lines may indirectly contribute to differences in muscle amino acid content through changes in protein metabolism.

In addition, considering that commercial chicken birds spend a significant proportion of their life in ovo, changing the ovo environment can permanently “program” the endocrine pathways of offspring [14]. And there were different metabolic patterns in F and L broilers [39]. We have also previously shown that the two lines subjected to restricted feeding differed in their effects on blood biochemical parameters and hormone levels in the offspring [8]. Restricted feeding in broiler breeders hens promoted protein synthesis in offspring, with lower protein synthesis in the F than that in the L [8]. In this study, only LR group breast muscle protein content in the growth period was higher than its normal group. The offspring of FR group was limited by hypoplastic skeletal muscles reflected by a high fat deposition content.

Hens exhibit chronic stress in the restricted feeding state, serum cortisol content is elevated [21]. In addition, mothers may transmit messages through the egg to change the characteristics of the offspring, such as greater adaptability in undernourished lives. There are complex synergistic effects of the yolk environment, in which yolk thyroid hormone and yolk testosterone as potential mediators of the physiological and morphological effects [40]. The restriction of feeding on broiler hens during the laying period affected the blood biochemical and hormone levels of the offspring during the embryonic period [41]. Maternally under high stress, offspring may exhibit high levels of testosterone [42], an anabolic hormone essential for muscle and bone development [43]. This may partially explain the elevated progeny intramuscular protein content of L hens under restricted feeding. Although the prenatal environment is important, restricted feeding handling only during egg production may not be sufficient to affect the phenotype of broilers [5]. In contrast to long-term malnutrition, nutritional deficiency of hens during egg production may only decrease the energy provided to reproduction. Phenotypic changes may be more readily transmitted to offspring by relevant hormonal and other factors if undernutrition begins from the chick age [44]. Different genders often possess distinct metabolic pathways, suggesting that maternal restriction of feeding differentially affects meat quality in offspring of different genders [21,45]. It is a negligence in this study that the different gender offspring were not separated.

More variability was observed in the thigh muscles compared to the breast muscles in the present study, possibly with a relationship to their myofiber types (not typed in this study). The pectoralis muscles contain only type IIb fibers, and thigh muscles that contain type I, IIa and IIb fiber [46]. The fastest growth of Type IIb fibers of hindlimb muscles [46] may be the reason for the more pronounced effect of the thigh muscles in this study. The dietary restriction of the mother made the muscle fibers of the peroneal longus of the 56-day-old broiler chicken thicker, especially the F. It could be that the leg muscle compensates for the need to support higher body weight in broilers, with too few muscle fibers requiring muscle thickening to provide greater support.

The changes in the amino acid composition of muscle, followed the flavor and taste of muscle [47,48], are very complex and still poorly understood. Branched chain amino acids (Leu, Val, and Ile) are associated with exercise capacity and promote muscle anabolism [49]. The differences in changes in muscle amino acid composition between the two lines suggested differences in muscle synthesis and metabolism between the two lines. In the breast muscles of F and L broilers, only the changes of Tyr content were consistent with those of thigh muscles observed, which may be worthy of further study. Considering that Tyr is a ketogenic sugar amino acid, the difference in Tyr content in breast muscle may be associated with the presence of crude protein and crude fat content in breast muscle during mid growth. The change of amino acid content was more embodied in thigh muscle, which may be related to the difference in the fiber type of thigh muscle.

It is commonly reported that many aspects of meat quality are related to muscle fiber characteristics [50]. In this experiment, the FR group was significantly lower than the FN group at the 48 h a* of day 56 thigh muscles. Possibly, FR group offspring thigh muscles had a larger proportion of type I muscle fibers because of higher oxygen consumption rate [51]. Zhu et al. (2012) [52] also reported the effects of maternal energy and protein levels on progeny meat color in Chinese yellow chickens. In this experiment, maternal feeding restriction did not affect the L*, b*, muscle pH, water-holding capacity, and tenderness of the offspring meat. Lv et al. (2012) [53] showed that the low nutrition level of the mother reduces the tenderness of the breast muscle of the offspring. Tenderness is affected by many factors such as the density of muscle fiber, intramuscular fat, and myosin [54]. Therefore, we inferred that the effect of maternal nutrition and the way of maternal nutrition restriction on muscle tenderness is inconsistent, and the factors involved may be complex and need further study. In addition, the sarcomere length and myofibril diameter of breast and thigh muscle of 56-day-old offspring was determined (Appendix A), and no significant difference was found. This indicated that restricted feeding in hens did not affect the state of myofibril contraction, relaxation, and disassembly during postmortem stiffness in the next generation of broilers.

Complex factors, factors affecting muscle development and growth metabolism such as individual genotypes and feeding procedures, management during slaughter, etc., all influence meat quality. The impact of maternal nutrition and altered fetal growth on the edible quality of meat is less than postnatal impacts, such as fattening at feeders or pastures, ante mortem management, and abattoir processing [55]. Similarly, prenatal effects on changes in carcass composition, such as fat, protein, and energy content, are similarly insignificant with respect to genotype and fattening in the rearing grounds [55].

In order to meet the requirements of meat quality and the best reproductive performance of broilers, special nutrition planning is needed for each genotype chicken breed if good production performance is maintained [56,57]. An innovative strategy to supplement nutrients into the egg via maternal feeding or in ovo-injections (‘in ovo feeding’) was described as an optimization of embryo nutrition supply [58].

## 5. Conclusions

Undernutrition may lead to growth deficiency and sarcopenia at birth in offspring and more deposition of fat in offspring with adequate nutrition after birth. However, during growth, most of the effects were improved by the broiler’s self-regulation abilities. Due to the different metabolic patterns, different broiler strains had different responses of meat traits to maternal nutrition interference. F mainly responded to intramuscular fat, while L focused on changes in intramuscular protein, such as changes in amino acid content. Further studies could be undertaken in the future and require more detailed systematic considerations including nutritional needs of broiler strains, chronic stress on the mother and its effect on offspring metabolism, offspring sex and growth trajectories, and fiber type of muscle.

## Figures and Tables

**Table 1 animals-11-02434-t001:** Composition and ingredients of the basal diets for laying hens and offspring.

Item	Laying Hens	Offspring (Weeks)		
		1–3	4–6	7–8
Corn (g kg^−1^)	628.8	570.0	612.6	658.8
Soybean meal (g kg^−1^)	235.0	326.0	300.0	270.0
Fish meal (g kg^−1^)	30.0	40.0	30.0	15.0
Oil (g kg^−1^)	-	30.0	25.0	25.0
NaCl (g kg^−1^)	3.0	2.0	1.0	0.8
CaHPO_4_ (g kg^−1^)	16.0	0.8	0.2	0.2
Limestone (g kg^−1^)	82.0	14.0	14.0	12.0
L-Lysine (g kg^−1^)	0.2	11.0	11.0	12.0
DL-methionine (g kg^−1^)	0.8	3.0	3.0	3.0
Choline chloride (g kg^−1^)	1.0	1.0	1.0	1.0
Vitamin premix (g kg^−1^) ^1^	0.2	0.2	0.2	0.2
Trace elements (g kg^−1^) ^2^	3.0	2.0	2.0	2.0
**Nutrient levels ^3^**				
Metabolizable energy (MJ kg^−1^)	11.80	13.00	13.03	13.19
Crude protein (g kg^−1^)	170.9	215.0	200.5	181.2
Lysine (g kg^−1^)	9.2	12.5	11.0	9.5
Methionine (g kg^−1^)	4.1	6.0	4.7	4.1
Calcium (g kg^−1^)	34.9	10.0	9.6	8.4
Available phosphorus (g kg^−1^)	4.9	4.6	4.3	4.0

^1^ Per kg of laying hens’ diet contained: VA 12000 IU; VD2400 IU; VE 30 IU; VK 1.5 mg; VB12 12 µg; biotin 0.2 mg; folate 1.2 mg; niacin 35 mg; pantothenic acid 12 mg; pyridoxine 4.5 mg; riboflavin 9 mg; thiamine 2.0 mg. Per kg of offspring broilers’ diet contained: VA 1500 IU; VD 3200 IU; VE 20 IU; VK 0.5 mg; VB_12_ 0.01 mg; biotin 0.15 mg; folate 0.55 mg; niacin 35 mg; pantothenic acid 10 mg; VB_6_ 3.5 mg; VB_2_ 3.6 mg; VB_1_ 1.8 mg. ^2^ Per kg of laying hens’ diet contained: cuprum 8 mg; iodine 1.0 mg; iron 80 mg; manganese 100 mg; selenium 0.30 mg; zinc 80 mg. Per kg of offspring broilers’ diet contained: cuprum 8 mg; iodine 0.35 mg; iron 80 mg; manganese 60 mg; selenium 0.15 mg; zinc 40 mg. ^3^ Calculated values.

**Table 2 animals-11-02434-t002:** Effect of feed intake on breast muscle color of 56-day-old offspring in broiler hens during laying period (n = 24).

Item	Line		Intake		Line × Intake				Pooled SEM	*p*-Value		
	L	F	NI	RI	LN	LR	FN	FR		Line	Intake	Line × Intake
**2 h**												
L*	47.98	48.83	48.42	48.39	48.34	47.63	48.51	49.15	1.200	0.333	0.965	0.442
a*	2.63 ^a^	1.79 ^b^	2.25	2.17	2.74	2.53	1.76	1.82	0.536	0.038	0.843	0.730
b*	7.88	7.37	7.60	7.65	8.06	7.70	7.14	7.60	1.152	0.540	0.951	0.620
**24 h**												
L*	49.31	51.07	50.29	50.09	48.86	49.77	51.73	50.41	1.210	0.054	0.812	0.209
a*	2.47 ^a^	1.58 ^b^	2.14	1.91	2.85	2.09	1.43	1.73	0.525	0.026	0.541	0.171
b*	7.49	7.64	7.46	7.67	7.43	7.54	7.48	7.79	1.019	0.836	0.775	0.895
**48 h**												
L*	49.79 ^b^	52.23 ^a^	51.04	50.98	49.76	49.82	52.32	52.13	1.335	0.018	0.948	0.893
a*	3.32 ^a^	2.13 ^b^	2.92	2.54	3.46	3.19	2.38	1.89	0.596	0.011	0.383	0.798
b*	8.60	8.57	8.78	8.39	8.71	8.48	8.85	8.29	1.190	0.972	0.646	0.849

Abbreviations: L, lean line; F, fat lean; NI, normal daily feed intake; RI, restricted daily feed intake; LN, lean line with normal daily feed intake; LR, lean line with restricted daily feed intake; FN, fat line with normal daily feed intake; FR, fat line with restricted daily feed intake; SEM, standard error of the mean; L*, lightness; a*, redness; b*, yellowness. ^a,b^ Different shoulder marks from same line were meant significant difference under each effect (*p* < 0.05).

**Table 3 animals-11-02434-t003:** Effect of feed intake on thigh muscle color of 56-day-old offspring in broiler hens during laying period (n = 24).

Item	Line		Intake		Line × Intake				Pooled SEM	*p*-Value		
	L	F	NI	RI	LN	LR	FN	FR		Line	Intake	Line × Intake
**2 h**												
L*	50.71	51.43	51.11	51.03	50.91	50.51	51.31	51.55	1.120	0.375	0.915	0.688
a*	5.63	7.56	7.34	5.86	5.68	5.58	8.99	6.14	1.358	0.062	0.145	0.171
b*	10.18	9.77	10.25	9.70	10.49	9.87	10.01	9.53	1.149	0.621	0.508	0.933
**24 h**												
L*	49.52	48.90	48.97	49.45	49.22	49.83	48.73	49.07	1.543	0.574	0.668	0.900
a*	6.55	6.31	6.78	6.08	7.12	5.98	6.44	6.18	1.299	0.795	0.451	0.639
b*	10.56	9.32	9.74	10.15	10.92	10.20	8.55	10.09	0.844	0.055	0.507	0.077
**48 h**												
L*	51.03 ^b^	53.19 ^a^	52.72	51.49	51.80	50.26	53.64	52.73	1.371	0.038	0.221	0.751
a*	6.98	7.24	7.32	6.90	6.13 ^b^	7.83 ^a,b^	8.50 ^a^	5.97 ^b^	1.250	0.776	0.645	0.031
b*	11.30	11.35	11.54	11.12	11.77	10.84	11.31	11.40	0.978	0.943	0.546	0.473

Abbreviations: L, lean line; F, fat lean; NI, normal daily feed intake; RI, restricted daily feed intake; LN, lean line with normal daily feed intake; LR, lean line with restricted daily feed intake; FN, fat line with normal daily feed intake; FR, fat line with restricted daily feed intake; SEM, standard error of the mean; L*, lightness; a*, redness; b*, yellowness. ^a,b^ Different shoulder marks from same line were meant significant difference under each effect (*p* < 0.05).

**Table 4 animals-11-02434-t004:** Effect of feed intake on pH value, press loss, drip loss, cooking loss, and shear force of muscle of 56-day-old offspring in broiler hens during laying period (n = 24).

Item	Line		Intake		Line × Intake				Pooled SEM	*p*-Value		
	L	F	NI	RI	LN	LR	FN	FR		Line	Intake	Line × Intake
**Breast muscle**												
0 h pH	5.84	5.74	5.80	5.79	5.86	5.82	5.73	5.76	0.072	0.069	0.898	0.503
24 h pH	5.90	5.78	5.87	5.80	5.88	5.91	5.86	5.70	0.083	0.057	0.246	0.120
**Thigh muscle**												
0 h pH	6.04	6.03	5.98	6.09	6.03	6.04	5.93	6.13	0.140	0.960	0.311	0.367
24 h pH	6.18	6.19	6.13	6.24	6.15	6.21	6.11	6.28	0.100	0.830	0.123	0.453
**Breast muscle**												
Press loss (%)	22.31	20.40	20.97	21.73	22.09	22.52	19.86	20.94	2.383	0.270	0.660	0.848
Drip loss (%)	1.72 ^a^	1.19 ^b^	1.27	1.64	1.49	1.95	1.06	1.33	0.262	0.013	0.064	0.621
Cook loss (%)	21.38	21.10	21.43	21.06	20.92	21.84	21.94	20.27	1.418	0.785	0.714	0.213
Shear force (N)	24.89	25.36	22.56	27.69	20.48	29.30	24.63	26.08	6.795	0.924	0.302	0.455

Abbreviations: L, lean line; F, fat lean; NI, normal daily feed intake; RI, restricted daily feed intake; LN, lean line with normal daily feed intake; LR, lean line with restricted daily feed intake; FN, fat line with normal daily feed intake; FR, fat line with restricted daily feed intake; SEM, standard error of the mean. ^a,b^ Different shoulder marks from same line were meant significant difference under each effect (*p* < 0.05).

**Table 5 animals-11-02434-t005:** Effect of feed intake on water, crude fat, and crude protein content of offspring muscle in broiler hens during laying period (n = 24).

Item	Line		Intake		Line × Intake				Pooled SEM	*p*-Value		
	L	F	NI	RI	LN	LR	FN	FR		Line	Intake	Line × Intake
**28-day-old**												
**Breast muscle**												
Water	69.42	68.68	71.38 ^a^	66.72 ^b^	71.71	67.13	71.05	66.30	2.318	0.654	0.007	0.959
Fat ^1^	6.89	6.94	6.15	7.68	7.26 ^a,b^	6.51 ^b,c^	5.03 ^c^	8.84 ^a^	1.083	0.951	0.055	0.006
Protein ^2^	87.22	86.88	86.72	87.37	85.58 ^b^	88.85 ^a^	87.86 ^a,b^	85.90 ^b^	1.298	0.715	0.484	0.007
**Thigh muscle**												
Water	71.40	69.67	70.26	70.80	71.33	71.47	69.19	70.14	3.304	0.463	0.818	0.864
Fat	25.01	25.97	24.02 ^b^	26.96 ^a^	23.41	26.61	24.63	27.31	1.969	0.493	0.040	0.853
Protein	67.42	66.56	67.99	65.99	68.09	66.74	67.88	65.25	1.793	0.505	0.124	0.615
**56-day-old**												
**Breast muscle**												
Water	73.22	73.73	74.53	72.41	74.30	72.14	74.76	72.69	1.782	0.690	0.108	0.974
Fat	3.41	3.50	3.71	3.30	3.95	2.87	3.48	3.52	0.614	0.835	0.252	0.218
Protein	88.68	89.05	88.97	88.76	88.74	88.62	89.19	88.91	1.307	0.692	0.829	0.929
**Thigh muscle**												
Water	76.48	76.30	76.55	76.23	76.94	76.01	76.15	76.44	0.686	0.712	0.521	0.227
Fat	10.81	11.95	11.15	11.61	11.45	10.16	10.85	13.06	1.504	0.297	0.668	0.120
Protein	78.99	76.80	77.88	77.91	78.10	79.88	77.65	75.94	1.847	0.114	0.981	0.202

Abbreviations: L, lean line; F, fat lean; NI, normal daily feed intake; RI, restricted daily feed intake; LN, lean line with normal daily feed intake; LR, lean line with restricted daily feed intake; FN, fat line with normal daily feed intake; FR, fat line with restricted daily feed intake; SEM, standard error of the mean. ^1,2^ Crude fat content and Crude protein content of air-dried sample. ^a–c^ Different shoulder marks from same line were meant significant difference under each effect (*p* < 0.05).

**Table 6 animals-11-02434-t006:** Effect of feed intake on amino acids contents of 56-day-old offspring breast muscle meat in broiler hens during the laying period (n = 24).

Item ^1^	Line		Intake		Line × Intake				Pooled SEM	*p*-Value		
	L	F	NI	RI	LN	LR	FN	FR		Line	Intake	Line × Intake
Asp	20.57	20.18	20.70	20.05	20.25	20.89	21.16	19.20	1.333	0.684	0.498	0.189
Thr	13.43	13.43	13.66	13.20	13.21	13.65	14.11	12.75	0.857	1.000	0.455	0.157
Ser	6.49	6.28	6.49	6.29	6.42	6.57	6.56	6.01	0.413	0.482	0.492	0.254
Glu	26.35	26.07	26.56	25.86	25.81	26.88	27.30	24.84	1.699	0.819	0.571	0.163
Gly	35.24	32.54	34.52	33.25	34.78	35.70	34.26	30.81	2.594	0.162	0.501	0.252
Ala	20.91	20.30	21.04	20.16	20.63	21.18	21.45	19.14	1.417	0.550	0.393	0.174
Cys	1.63	1.73	1.74	1.63	1.71	1.56	1.76	1.70	0.118	0.252	0.243	0.625
Val	11.85	12.04	12.15	11.74	11.72	12.00	12.58	11.50	0.787	0.740	0.474	0.246
Met	5.05 ^b^	6.01 ^a^	5.73	5.32	5.17	4.93	6.29	5.72	0.382	0.003	0.155	0.550
Ile	12.69	13.62	13.38	12.94	12.57	12.81	14.18	13.06	0.821	0.131	0.460	0.258
Leu	20.22 ^b^	23.23 ^a^	22.04	21.41	20.03	20.41	24.06	22.41	1.374	0.007	0.523	0.310
Tyr	5.47 ^a^	3.11 ^b^	4.36	4.21	5.04 ^b^	5.89 ^a^	3.68 ^c^	2.54 ^d^	0.265	<0.001	0.450	<0.001
Phe	14.55 ^b^	16.19 ^a^	15.66	15.09	14.28	14.82	17.04	15.35	0.923	0.023	0.394	0.107
Lys	23.50	22.59	23.39	22.70	23.12	23.88	23.65	21.52	1.503	0.402	0.528	0.194
His	10.86	9.84	10.64	10.06	10.67	11.06	10.62	9.07	0.739	0.070	0.285	0.082
Arg	14.86	15.25	15.49	14.62	15.10	14.63	15.88	14.62	1.132	0.637	0.297	0.634
Pro	17.22	15.57	16.62	16.17	16.87	17.56	16.36	14.79	1.221	0.076	0.617	0.211

Abbreviations: L, lean line; F, fat lean; NI, normal daily feed intake; RI, restricted daily feed intake; LN, lean line with normal daily feed intake; LR, lean line with restricted daily feed intake; FN, fat line with normal daily feed intake; FR, fat line with restricted daily feed intake; SEM, standard error of the mean; Asp, aspartic acid; Thr, Threonine; Ser, Serine; Glu, Glutamic acid; Gly, Glycine; Ala, Alanine; Cys, Cysteine; Val, Valine; Met, Methionine; Ile, Isoleucine; Leu, Leucine; Tyr, Tyrosine; Phe, Phenylalanine; Lys, lysine; His, Histidine; Arg, Arginine; Pro, Proline. ^1^ the unit for amino acid contents was ug/mg. ^a–d^ Different shoulder marks from same line were meant significant difference under each effect (*p* < 0.05).

**Table 7 animals-11-02434-t007:** Effect of feed intake on amino acids contents of 56-day-old offspring thigh muscle in broiler hens during laying period (n = 24).

Item ^1^	Line		Intake		Line × Intake				Pooled SEM	*p*-Value		
	L	F	NI	RI	LN	LR	FN	FR		Line	Intake	Line × Intake
Asp	18.23	18.53	18.33	18.43	17.38 ^b^	19.09 ^a,b^	19.27 ^a^	17.78 ^a,b^	0.932	0.660	0.872	0.025
Thr	12.15	12.13	12.10	12.17	11.56 ^b^	12.73 ^a^	12.64 ^a,b^	11.62 ^a,b^	0.599	0.970	0.860	0.017
Ser	6.14	6.08	6.06	6.15	5.82 ^b^	6.46 ^a^	6.30 ^a,b^	5.85 ^a,b^	0.294	0.763	0.679	0.017
Glu	25.23	25.00	25.09	25.14	24.14 ^b^	26.32 ^a^	26.05 ^a,b^	23.96 ^b^	1.257	0.803	0.958	0.026
Gly	32.58	32.64	32.28	32.94	30.56 ^b^	34.61 ^a^	34.00 ^a,b^	31.28 ^b^	2.051	0.969	0.653	0.030
Ala	18.54	18.62	18.52	18.64	17.59 ^b^	19.49 ^a^	19.45 ^a^	17.79 ^a,b^	0.999	0.916	0.867	0.020
Cys	1.67	1.67	1.66	1.68	1.61	1.73	1.70	1.64	0.087	0.983	0.656	0.175
Val	10.50	10.45	10.46	10.49	9.92 ^b^	11.08 ^a^	10.99 ^a^	9.91 ^b^	0.459	0.884	0.911	0.003
Met	4.76	4.86	4.89	4.73	4.80	4.72	4.97	4.75	0.291	0.626	0.456	0.729
Ile	11.39	11.47	11.40	11.46	10.78 ^b^	11.99 ^a^	12.01 ^a^	10.93 ^b^	0.476	0.818	0.849	0.003
Leu	18.83	18.56	18.57	18.82	17.69 ^b^	19.97 ^a^	19.45 ^a^	17.67 ^b^	0.729	0.604	0.637	0.001
Tyr	4.84 ^a^	4.35 ^b^	4.54	4.65	4.24 ^c^	5.45 ^a^	4.84 ^b^	3.85 ^c^	0.249	0.013	0.537	<0.001
Phe	11.71	11.87	11.74	11.84	10.97 ^b^	12.45 ^a^	12.51 ^a^	11.22 ^b^	0.422	0.610	0.754	<0.001
Lys	21.35	21.67	21.42	21.60	20.20 ^b^	22.49 ^a,b^	22.63 ^a^	20.70 ^a,b^	1.143	0.696	0.826	0.017
His	7.70	7.91	7.97	7.65	7.57	7.84	8.36	7.46	0.400	0.469	0.273	0.051
Arg	13.98	14.06	13.86	14.18	13.02 ^b^	14.95 ^a^	14.70 ^a,b^	13.42 ^a,b^	0.967	0.907	0.641	0.029
Pro	15.66	15.47	15.40	15.73	14.88	16.44	15.92	15.02	1.097	0.816	0.676	0.128

Abbreviations: L, lean line; F, fat lean; NI, normal daily feed intake; RI, restricted daily feed intake; LN, lean line with normal daily feed intake; LR, lean line with restricted daily feed intake; FN, fat line with normal daily feed intake; FR, fat line with restricted daily feed intake; SEM, standard error of the mean; Asp, aspartic acid; Thr, Threonine; Ser, Serine; Glu, Glutamic acid; Gly, Glycine; Ala, Alanine; Cys, Cysteine; Val, Valine; Met, Methionine; Ile, Isoleucine; Leu, Leucine; Tyr, Tyrosine; Phe, Phenylalanine; Lys, lysine; His, Histidine; Arg, Arginine; Pro, Proline. ^1^ the unit for amino acid contents was μg/mg. ^a–c^ Different shoulder marks from same line were meant significant difference under each effect (*p* < 0.05).

**Table 8 animals-11-02434-t008:** Effect of feed intake on diameter and density of offspring muscle fiber in broiler hens during laying period (n = 24).

Item	Line		Intake		Line × Intake				Pooled SEM	*p*-Value		
	L	F	NI	RI	LN	LR	FN	FR		Line	Intake	Line × Intake
**1-d** **ay** **-old**												
**Diameter (um)**												
Breast muscle	8.48	8.31	8.43	8.36	8.14 ^a,b^	8.82 ^a^	8.72 ^a,b^	7.90 ^b^	0.490	0.632	0.848	0.044
Thigh muscle	10.60	10.13	10.67	10.06	10.86	10.34	10.49	9.78	0.521	0.227	0.119	0.801
**Density (per mm^2^)**												
Breast muscle	7331.2	7356.1	7369.4	7317.9	8115.9 ^a^	6546.4 ^b^	6622.9 ^a,b^	8089.3 ^a,b^	889	0.969	0.935	0.025
Thigh muscle	5260.6	5060.5	4842.6	5478.5	4649.4	5871.7	5035.8	5085.2	635.3	0.661	0.172	0.207
**56-day-old**												
**Diameter (um)**												
Breast muscle	43.82	40.36	42.55	41.63	45.86	41.77	39.24	41.48	2.472	0.067	0.605	0.091
Peroneus longus	46.77	45.98	43.93 ^b^	48.82 ^a^	45.95	47.59	41.92	50.04	2.544	0.665	0.013	0.087
Gastrocnemius	47.40	45.32	47.34	45.38	47.05	47.76	47.63	43.01	3.281	0.379	0.409	0.265
**Density (per mm^2^)**												
Breast muscle	293.8	357.9	340.9	310.8	286.7	300.9	395.1	320.7	46.08	0.068	0.371	0.194
Peroneus longus	277.1	287.7	311.9 ^a^	252.9 ^b^	284.3	269.8	339.4	235.9	32.52	0.649	0.019	0.067
Gastrocnemius	250.9 ^b^	322.2 ^a^	272.1	301.0	254.4	247.4	289.7	354.7	42.45	0.031	0.351	0.249

Abbreviations: L, lean line; F, fat lean; NI, normal daily feed intake; RI, restricted daily feed intake; LN, lean line with normal daily feed intake; LR, lean line with restricted daily feed intake; FN, fat line with normal daily feed intake; FR, fat line with restricted daily feed intake; SEM, standard error of the mean. ^a,b^ Different shoulder marks from same line were meant significant difference under each effect (*p* < 0.05).

## Data Availability

The data presented in this study are available in the present article and Appendix A and are shared with consent and in accordance with all authors.

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
