# Peer review of "Effects of Nutritional Restriction during Laying Period of Fat and Lean Line Broiler Breeder Hens on Meat Quality Traits of Offspring"

_animals, 2021, doi:10.3390/ani11082434_

Round 1

Reviewer 1 Report

Thank you for your effort to clarify and explain better, I woudl like to see also in results and discussion some division chapters  among different subjects (e.g. muscle, bones, fiber) just to simplify the reading, but it's just a stylistic comment.

Reviewer 2 Report

As the manuscript has been revised well, I think that it can be published in the journal. 

This manuscript is a resubmission of an earlier submission. The following is a list of the peer review reports and author responses from that submission.

Round 1

Reviewer 1 Report

Materials and methods must be improved, since two different experiments are carried out it is necessary to better ditinguish them with different subtitles separating laying hens and broilers.  Pay attentio to paragrahps caoming from the editors suggestion and are not part of the study.  Explain better the treatment groups and number of treatments that seems not correspond and the difference between L and LN and F and FN.

The topic is not original, restricted feeding vs ad libitum feeding in broiler breeders has been discussed for many years, however the study add the performance analysis of the offspring, that adds something to the previous studies.

I think that experimental and statistic parts sound correct but the structure of the study has to be changed. The authors should divide the experiments in chapters and sub-chapters in every part of the article. Material and method are not clearly explained in terms of group division, letter and treatment. The two experiments should be divided in different chapters and discussed with more consequentiality. Being the study full of data and results, it is necessary to give a logic and rigorous order to facilitate the reader comprehension. I suggest to refer to other studies with more than one experiment to take a cue on how to improve the article structure.

Reviewer 2 Report

Overall, this is a good manuscript for publication. Some areas could be discussed in more detail.

Materials and Methods: 

2.1. first paragraph, last sentence could be stated more clearly.

2.2. You said that at 40 weeks AI was used. Please provide more details.

2.2. There is a statement after Table 1 describing what the Materials and Methods should contain. That should be removed.

2.3. It would be helpful to describe how the drip loss, cook loss, and shear force were determined instead of only naming the equipment used.

2.3. 4th paragraph- the first sentence is incomplete

2.3. It is mentioned that samples were oven dried for crude protein, crude fat, and amino acid, then the next sentence air dried samples were used to  determine crude protein and fat content. Please explain more clearly. 

2.4. You used Randomized Complete Block Design, but it was not clear what was blocked.

Results

3. 'press loss' is mentioned twice in the sentence.

Reviewer 3 Report

This manuscript includes interesting results on effects of nutritional restriction during laying period of two different broiler breeder hens on meat quality traits of offspring chicken. However, some parts of manuscript should be corrected for publication. Please refer to an attached file. 
